# RCE-IFE: recursive cluster elimination with intra-cluster feature elimination

Cihan Kuzudisli[1,2], Burcu Bakir-Gungor[3], Bahjat Qaqish[4] and
Malik Yousef[5,6]

[1] Department of Computer Engineering, Faculty of Engineering, Hasan Kalyoncu University,
Gaziantep, Turkey
[2] Department of Electrical and Computer Engineering, Abdullah Gul University, Kayseri, Turkey
[3] Department of Computer Engineering, Faculty of Engineering, Abdullah Gul University, Kayseri,
Turkey
[4] Department of Biostatistics, University of North Carolina at Chapel Hill, North Carolina,
Chapel Hill, United States
[5] Department of Information Systems, Zefat Academic College, Zefat, Israel
[6] Galilee Digital Health Research Center, Zefat Academic College, Zefat, Israel

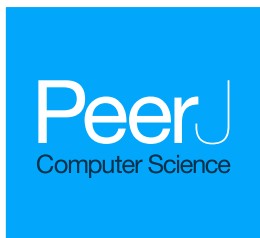

Corresponding author
Malik Yousef,
malik.yousef@gmail.com

## ABSTRACT

The computational and interpretational difficulties caused by the ever-increasing
dimensionality of biological data generated by new technologies pose a significant
challenge. Feature selection (FS) methods aim to reduce the dimension, and feature
grouping has emerged as a foundation for FS techniques that seek to detect strong
correlations among features and identify irrelevant features. In this work, we propose
the Recursive Cluster Elimination with Intra-Cluster Feature Elimination (RCE-IFE)
method that utilizes feature grouping and iterates grouping and elimination steps in a
supervised context. We assess dimensionality reduction and discriminatory
capabilities of RCE-IFE on various high-dimensional datasets from different biological
domains. For a set of gene expression, microRNA (miRNA) expression, and
methylation datasets, the performance of RCE-IFE is comparatively evaluated with
RCE-IFE-SVM (the SVM-adapted version of RCE-IFE) and SVM-RCE. On average,
RCE-IFE attains an area under the curve (AUC) of 0.85 among tested expression
datasets with the fewest features and the shortest running time, while RCE-IFE-SVM
(the SVM-adapted version of RCE-IFE) and SVM-RCE achieve similar AUCs of 0.84
and 0.83, respectively. RCE-IFE and SVM-RCE yield AUCs of 0.79 and 0.68,
respectively when averaged over seven different metagenomics datasets, with RCE-IFE
significantly reducing feature subsets. Furthermore, RCE-IFE surpasses several state-
of-the-art FS methods, such as Minimum Redundancy Maximum Relevance
(MRMR), Fast Correlation-Based Filter (FCBF), Information Gain (IG), Conditional
Mutual Information Maximization (CMIM), SelectKBest (SKB), and eXtreme
Gradient Boosting (XGBoost), obtaining an average AUC of 0.76 on five gene
expression datasets. Compared with a similar tool, Multi-stage, RCE-IFE gives a
similar average accuracy rate of 89.27% using fewer features on four cancer-related
datasets. The comparability of RCE-IFE is also verified with other biological domain
knowledge-based Grouping-Scoring-Modeling (G-S-M) tools, including
mirGediNET, 3Mint, and miRcorrNet. Additionally, the biological relevance of the
selected features by RCE-IFE is evaluated. The proposed method also exhibits high
consistency in terms of the selected features across multiple runs. Our experimental
findings imply that RCE-IFE provides robust classifier performance and significantly
reduces feature size while maintaining feature relevance and consistency.

**How to cite this article** Kuzudisli C, Bakir-Gungor B, Qaqish B, Yousef M. 2025. RCE-IFE: recursive cluster elimination with intra-cluster
feature elimination. **PeerJ Comput. Sci.** 11:e2528 DOI 10.7717/peerj-cs.2528

## INTRODUCTION

Recent advances in next generation sequencing and DNA microarray technologies allow scientists to easily access substantial amounts of gene expression data. The analysis of these data enables the discovery of coexpression patterns among genes and their connections to disease in order to achieve deeper insights into the molecular mechanisms of disease development, improve diagnosis, and develop more effective treatment plans (*Rosati et al., 2024*). Particularly, the identification of disease-associated genes, namely biomarkers, is essential for prognostic prediction, early diagnosis, and drug discovery (*Yousef, Kumar & Bakir-Gungor, 2020*). Gene expression data typically comes with a relatively small number of samples accompanied by a huge number of genes (*Kuzudisli et al., 2024*). This characteristic makes data processing and analysis a challenging task and it is considered a hot topic in the field of bioinformatics (*Clark & Lillard, 2024*). Feature selection (FS) is an effective dimensionality reduction technique and, unlike other reduction techniques such as principal component analysis, it does not alter the original representation of the features but essentially opts for a subset of the original feature set. Hence, FS preserves the semantics of the features in their original form, allowing domain experts to interpret the data. FS is a prerequisite in the realm of bioinformatics due to the existence of redundancy and noise in the biological data (*Cai et al., 2018*). In the context of feature subset selection, FS methods are classified as filter, wrapper, and embedded methods (*Gonzalez-Lopez, Ventura & Cano, 2020*). Filter methods utilize statistical measures for feature assessment and are independent of any classifier. In wrapper methods, FS is carried out using a learning algorithm along with a certain search strategy. Embedded methods accomplish FS and model construction simultaneously and are specific to an induction algorithm like wrapper methods. However, they have less computational cost than wrapper methods. Later, ensemble and hybrid methods were derived based on these three methods (*Pudjihartono et al., 2022*; *Kuzudisli et al., 2023*). Many FS techniques are available in the literature and are widely used in gene subset extraction and disease classification (*Perscheid, 2021*).

Recursive cluster elimination based on support vector machine (SVM-RCE), proposed by *Yousef et al. (2007)*, introduced the term *recursive cluster elimination* into the literature, and this approach predominated over support vector machines with recursive feature elimination (SVM-RFE) (*Guyon et al., 2002*), which was widely accepted as an effective approach in the field. The superiority of SVM-RCE stems from the consideration of feature (*i.e.*, gene) clusters instead of individual features in the classification task. With the advent of SVM-RCE, similar approaches emerged to perform cluster selection rather than individual feature selection. *Tang et al. (2008)* group features into a fixed number of clusters and eliminate lower-ranked features in clusters. They repeat these steps until the remaining features are lowered to a predefined threshold. *Du et al. (2013)* proposed Multi-stage, where features are first grouped into clusters, the clusters are ranked by

backward elimination using SVM-RFE, and those with a single feature are discarded to a certain extent. Subsequently, the features in each cluster are ranked by SVM-RFE, including clusters whose rank is higher than that cluster, and a certain proportion of features is removed. Finally, all remaining features from the clusters are gathered, and SVM-RFE is employed once again to create the final feature subset. *Huang (2021)* presented RBF-RCE, utilizing RBF (*Wang et al., 2016*) to calculate feature importance. The features are then divided into clusters, and the highest importance value among the features in a cluster is assigned as the score of that cluster. RCE is then applied to remove clusters until a pre-specified number of clusters remains. Lastly, feature removal is carried out in the remaining clusters based on feature importance.

SVM-RCE, as originally described or with some variations, was used in numerous studies, including those related to neuroimaging (*Palaniyappan et al., 2019*; *Lanka et al., 2020*; *Karunakaran, Babiker Hamdan & Sathish, 2020*). *Weis, Visco & Faulon (2008)* modified SVM-RCE by applying SVM on all clusters and dropping clusters one at a time. In their approach, the cluster whose removal maximizes accuracy is excluded, and this process is repeated to obtain a subset of clusters with high predictive power. To reduce execution time, *Luo et al. (2011)* trained SVM using all clusters together and selected the highest absolute SVM feature weight within a cluster as a cluster score rather than relying on cross-validation. In *Rangaprakash et al. (2017)*, SVM-RCE was used to classify individuals as post-traumatic stress disorder (PTSD), post-concussion syndrome (PCS) + PTSD, or controls. *Jin et al. (2017)* adopted an SVM-RCE-like approach to discriminate between individuals with PTSD and healthy controls in their study on brain connectivity. *Zhao, Wang & Chen (2017)* compared SVM-RCE with other tools to detect expression profiles for identifying microRNAs associated with venous metastasis in hepatocellular carcinoma. *Chaitra, Vijaya & Deshpande (2020)* employed SVM-RCE to evaluate the classification performance of different feature sets in biomarker-based detection of autism spectrum disorder (ASD). Furthermore, the merit of the original SVM-RCE has contributed to the development of an approach called G-S-M (*Yousef et al., 2024*) that integrates biological prior knowledge. The G-S-M approach forms the basis for developing tools such as maTE (*Yousef, Abdallah & Allmer, 2019*), PriPath (*Yousef et al., 2023*), GediNET (*Qumsiyeh, Showe & Yousef, 2022*), miRcorrNet (*Yousef et al., 2021*), 3Mint (*Unlu Yazici et al., 2023*), GeNetOntology (*Ersoz, Bakir-Gungor & Yousef, 2023*), TextNetTopics (*Yousef & Voskergian, 2022*), TextNetTopics Pro (*Voskergian, Bakir-Gungor & Yousef, 2023*), microBiomeGSM (*Bakir-Gungor et al., 2023*), miRGediNET (*Qumsiyeh, Salah & Yousef, 2023*), miRdisNET (*Jabeer et al., 2023*), miRModuleNet (*Yousef, Goy & Bakir-Gungor, 2022*), CogNet (*Yousef, Ülgen & Uğur Sezerman, 2021*), and AMP-GSM (*Söylemez, Yousef & Bakir-Gungor, 2023*), which integrate biological networks and prior knowledge to provide a comprehensive understanding of genetic interactions.

In this article, we extend SVM-RCE by integrating feature elimination within surviving clusters at each step of cluster reduction. Through this newly added phase, less contributing features in each cluster are excluded, leading to deeper dimension reduction and also improvements in feature subset quality, classification performance, and running

time. The proposed Recursive Cluster Elimination with Intra-cluster Feature Elimination (RCE-IFE) tool can tackle large-scale datasets. We evaluate the dimension reduction capability and predictive power of RCE-IFE on various datasets. For a combination of gene expression, microRNA (miRNA) expression, and methylation datasets, RCE-IFE achieves an average area under the curve (AUC) of 0.85 using the least number of features with the least execution time, while RCE-IFE-SVM (SVM-adapted version of RCE-IFE) and SVM-RCE generate similar average AUCs of 0.84 and 0.83, respectively. When tested on seven different metagenomics datasets, RCE-IFE and SVM-RCE yields an average AUC of 0.79 and 0.68 respectively, while RCE-IFE providing a remarkable reduction in feature subsets. Moreover, RCE-IFE predominates among several state-of-the-art FS methods, including MRMR, FCBF, IG, CMIM, SKB, and XGBoost, by providing an average AUC of 0.76 on five gene expression datasets. In comparison with Multi-stage algorithm, RCE-IFE yields a similar accuracy rate of 89.27%, averaged over four cancer-related datasets using a smaller number of features. Additionally, we have comparatively evaluated RCE-IFE with other biological domain knowledge-based Grouping-Scoring-Modeling (G-S-M) tools (mirGediNET, 3Mint, and miRcorrNet) and conducted biological validation of the selected features. We further show the high consistency of the features selected by RCE-IFE across multiple runs. Overall, *via* testing through diverse biological datasets concerning various diseases, the experimental findings show the effectiveness of the proposed RCE-IFE method with respect to predictive power, reduced feature subset selection, feature relevancy, and consistency of selected features across different runs.

The remainder of the article is organized as follows: The Material and Methods section presents the main characteristics of datasets used in the study and provides a detailed explanation of the proposed approach. The Results and Discussion section presents the findings of various experimental tests, explains the results of comparative performance evaluation, and discusses the biological validity of the selected features. Finally, the Conclusion section concludes with our main insights and possible directions for future research. Portions of this text were previously published as part of a preprint (https://www.biorxiv.org/content/10.1101/2024.02.28.580487v1).

# MATERIALS AND METHODS

## Datasets

The datasets used in this article cover a broad range of biological domains and disease types. Table 1 summarizes the utilized 20 datasets on gene expression, methylation, and miRNA accessed from GEO (*Barrett et al., 2012*) and TCGA (*Tomczak, Czerwińska & Wiznerowicz, 2015*) databases. While gene expression datasets are available at GEO; TCGA-BLCA.methylation450, TCGA-BLCA.mirna and TCGA-BRCA.methylation450 datasets can be obtained from UCSC Xena repository (https://xenabrowser.net/datapages/) (*Goldman et al., 2020*). TCGA-BLCA.methylation450 and TCGA-BLCA.mirna datasets are accessible in the GDC TCGA Bladder Cancer (BLCA) cohort. TCGA-BRCA.methylation450 dataset can be found in the GDC TCGA Breast Cancer (BRCA) cohort.

Table 2 describes seven metagenomics datasets involving colorectal cancer (CRC), inflammatory bowel disease (IBD), Inflammatory Bowel Disease Multi-omics Database

**Table 1  Basic information about the gene expression, miRNA, and methylation datasets.**

| Dataset | #Samples | Label: count | #Features | Disease type |
|---|---|---|---|---|
| GDS1962 | 180 | Pos: 157, Neg: 23 | 54,613 | Glioma |
| GDS2519 | 105 | Pos: 50, Neg: 55 | 22,283 | Parkinson's disease |
| GDS2547 | 164 | Pos: 75, Neg: 89 | 12,646 | Prostate cancer |
| GDS2609 | 22 | Pos: 12, Neg: 10 | 54,635 | Colorectal cancer |
| GDS3268 | 200 | Pos: 129, Neg: 71 | 44,289 | Ulcerative colitis |
| GDS3646 | 132 | Pos: 110, Neg: 22 | 22,185 | Celiac disease |
| GDS3794 | 33 | Pos: 18, Neg: 15 | 48,702 | Rheumatoid arthritis |
| GDS3837 | 120 | Pos: 60, Neg: 60 | 30,622 | Non-small cell lung carcinoma |
| GDS3874 | 117 | Pos: 93, Neg: 24 | 22,284 | Diabetes |
| GDS3875 | 117 | Pos: 93, Neg: 24 | 22,645 | Diabetes |
| GDS3929 | 64 | Pos: 45, Neg: 19 | 24,527 | Tobacco smoke-related defects |
| GDS4228 | 166 | Pos: 147, Neg: 19 | 4,776 | Human immunodeficiency virus |
| GDS4824 | 21 | Pos: 13, Neg: 8 | 54,635 | Prostate cancer |
| GDS5037 | 108 | Pos: 88, Neg: 20 | 41,000 | Severe asthma |
| GDS5093 | 56 | Pos: 47, Neg: 9 | 54,613 | Acute dengue |
| GDS5499 | 140 | Pos: 99, Neg: 41 | 48,803 | Pulmonary hypertension |
| GSE157103 | 126 | Pos: 100, Neg: 26 | 19,472 | Coronavirus disease 2019 |
| TCGA-BLCA.methylation450 | 425 | Pos: 408, Neg: 17 | 20,623 | Bladder cancer |
| TCGA-BLCA.mirna | 425 | Pos: 408, Neg: 17 | 1,881 | Bladder cancer |
| TCGA-BRCA.methylation450 | 124 | Pos: 36, Neg: 88 | 15,770 | Breast cancer |

**Table 2  Basic information about the metagenomics datasets.**

| Dataset | #Samples | Label: count | #Features | Disease type | Reference |
|---|---|---|---|---|---|
| CRC_enzyme | 1,262 | Pos: 600, Neg: 662 | 2,875 | Colorectal cancer | Beghini et al. (2021) |
| CRC_pathway | 1,262 | Pos: 600, Neg: 662 | 549 | Colorectal cancer | Beghini et al. (2021) |
| CRC_species | 1,262 | Pos: 600, Neg: 662 | 917 | Colorectal cancer | Beghini et al. (2021) |
| CRC_species_II | 108 | Pos: 60, Neg: 48 | 528 | Colorectal cancer | Zeller et al. (2014) |
| IBD | 382 | Pos: 148, Neg: 234 | 534 | Inflammatory bowel disease | MetaHIT Consortium et al. (2010) |
| IBDMDB | 1,638 | Pos: 1209, Neg: 429 | 578 | Inflammatory bowel disease | Beghini et al. (2021) |
| T2D | 290 | Pos: 155, Neg: 135 | 587 | Type 2 diabetes | Qin et al. (2012) |

(IBDMDB), and Type 2 diabetes (T2D). CRC_species, CRC_pathway, and CRC_enzyme datasets were created and presented as a Supplemental Material in *Beghini et al.*'s *(2021)* study (*Beghini et al., 2021*). The IBD dataset is obtained from the European Nucleotide Archive (ENA) database with accession number ERA000116. The IBDMDB dataset is available at Sequence Read Archive (SRA) with accession number PRJNA398089. The T2D dataset is provided by NCBI Sequence Read Archive with accession numbers SRA045646 and SRA050230. The CRC_species_II dataset can be accessed from the ENA database with accession number PRJEB6070.

**Table 3 Basic information about the cancer-related datasets.**

| Dataset | #Samples | Label: count | #Genes | Reference |
|---------|----------|--------------|--------|-----------|
| Breast | 117 | Positive: 28, Negative: 89 | 11,885 | *van't Veer et al. (2002)* |
| DLBCL | 77 | DLBCL: 58, FL: 19 | 7,129 | *Shipp et al. (2002)* |
| Leukemia | 72 | ALL: 47, AML: 25 | 7,129 | *Golub et al. (1999)* |
| Prostate | 102 | Tumor: 52, Normal: 50 | 12,600 | *Singh et al. (2002)* |

**Note:**
ALL, acute lymphoblastic leukemia; AML, acute myeloid leukemia; DLBCL, diffuse large B-cell lymphoma; FL, follicular lymphoma.

Lastly, four publicly available cancer-related datasets are described in Table 3. The leukemia dataset can be downloaded at https://www.openintro.org/data/index.php?data=golub. The prostate dataset can be extracted from the R package SIS (details are available at https://cran.r-project.org/web/packages/SIS/SIS.pdf). The breast dataset, available in the Breast Cancer (*van't Veer et al., 2002*) cohort, can be accessed through the UCSC Xena platform (*Goldman et al., 2020*) at https://xenabrowser.net/datapages/. Finally, the DLBCL dataset can be downloaded at https://github.com/ramhiser/datamicroarray.

## Proposed algorithm

The proposed algorithm, called RCE-IFE, follows an iterative process in which each iteration involves the following two elimination steps: firstly, weakly-scoring clusters are removed; secondly, intra-cluster elimination is performed for the low-scoring features in the surviving clusters. Let $M$ be the $K \times 1$ vector of cluster numbers in descending order. The number of clusters in the $i$-th iteration is $M_i$, where $M_i < \ldots < M_K$ represents a user-supplied decreasing sequence. The user also controls the rate of intra-cluster feature elimination. The data, consisting of $p$ samples and $q$ features, is organized in a $p \times q$ matrix, $D$, accompanied by a vector of class labels $S$. To begin with, we create a training set and a test set by randomly splitting the samples, *i.e.*, the rows of $D$, into $D_{train}$ ($r_{train}\%$) and $D_{test}$ ($r_{test}\%$). A two-sided t-test is applied to each feature in $D_{train}$ to select the top 1,000 features with the smallest p-values. Let $F_0$ denote the set of filtered features. The subsequent operations are performed recursively. In the $i$-th iteration, $i = 1, \ldots, K\text{-}1$, K-means is used to group $F_{i-1}$ into $M_i$ clusters $C_i = \{C_{i1}, C_{i2}, \ldots, C_{iM_i}\}$. Next, each $C_{ij} \in C_i$ is assigned the mean of t estimates of classification accuracy. Algorithm 1 presents the pseudocode for assigning a score to a cluster. Clusters with the lowest $M_i - M_{i+1}$ scores are eliminated, leaving $M_{i+1}$ clusters in $C_i$. Within each cluster, intra-cluster feature importance weights are estimated by Random Forest (RF) algorithm, and the lowest-ranked $f\%$ of features are eliminated. Algorithm 2 gives the pseudocode for scoring the features in a surviving cluster. Finally, $D_{train}$ and $D_{test}$ are updated to contain only the surviving features, and the performance of the learned model is computed using RF. To give a brief example, if $M = \{100, 90, 70\}$, in the first iteration, the algorithm eliminates $100 - 90 = 10$ clusters, performs intra-cluster feature elimination on $f\%$ of the features in the 90 surviving clusters, and outputs performance results based on 90 clusters. In the

---

**Algorithm 1** Cluster scoring—*Score* $(C, D_{train}, s, t)$.

$C$ = a set of feature identifiers

$D_{train}$ = training set with a certain portion of the original sample set

$s$ = set of class labels corresponding to samples in $D_{train}$

$D_{train_C}$ = subset of $D_{train}$ containing features only in C with the corresponding sample values

$t$ = number of partitions

$acc$ = [] an empty array of length $t$

Step 1: for $i = 1 : t$ do

Step 2: Split the samples randomly in $D_{train_C}$ into $d_{train}$ (70%) and $d_{test}$ (30%) and, correspondingly, $s$ into $s_{train}$ and $s_{test}$

Step 3: Train RF Learner using $d_{train}$ and $s_{train}$

Step 4: $acc[i]$ = test RF on $d_{test}$ and $s_{test}$-compute accuracy of the model

Step 5: end

Step 6: return mean $(acc)$

---

**Algorithm 2** Intra-cluster feature scoring—*Importance* $(C, D_{train}, s)$.

$C$ = a set of feature identifiers

$D_{train}$ = training set with a certain portion of the original sample set

$s$ = vector of class labels corresponding to samples in $D_{train}$

$D_{train_C}$ = subset of $D_{train}$ containing features only in C with the corresponding sample values

Step 1: Train RF Learner using $D_{train_C}$ and $s$.

Step 2: Get measures of feature importance.

---

second iteration, the algorithm eliminates 90 − 70 = 20 clusters, conducts intra-cluster feature elimination in the 70 surviving clusters, and generates performance results based on 70 clusters. The algorithm would then halt, yielding performance measures for both the 90 and 70 clusters. The full steps of RCE-IFE are given in Algorithm 3.

The main steps in RCE-IFE are: Grouping (G), Scoring (S), Intra-cluster Feature Elimination (IFE), and Modeling (M). The workflow of RCE-IFE is illustrated in Fig. 1, and details of each step are explained in the next subsection.

**Grouping (G) step:** The first step involves utilizing the training dataset to group the currently active genes into a predetermined number of clusters. In our tool, K-means is employed, though other methods could also be applied. Let $C = \{c_1, c_2, \ldots, c_k\}$ represent the set of feature clusters. For each cluster $c_j$, a two-class subdataset is constructed, containing all features in $c_j$, along with the corresponding sample values and class labels. As a result, $k$ two-class subdatasets are generated in this step. Figure 2 illustrates this process, showing a training set of ten samples (rows) and ten genes (columns). In this example, four gene clusters (upper right) are formed, leading to the generation of four two-class subdatasets, which are then fed into the S step for scoring, ranking, and elimination.

---

**Algorithm 3   RCE-IFE.**

**Input:** $D = p \times q$ matrix representing $p$ samples and $q$ features.

$S$ is a $p \times 1$ vector of class labels. Each $S_j$ is either pos or neg

$M$ is a $K \times 1$ vector of numbers of clusters arranged in decreasing order

$r_{train}$: percentage of samples in the training set. $r_{test}$ is defined to be $100 - r_{train}$

$f$ = percentage of intra-cluster features to be removed

**Output:** The performance results for different numbers of clusters

Step 1: Partition the rows of $D$ randomly into $D_{train}$ (90%) and $D_{test}$ (10%)

Step 2: Apply the t-test to compare the two classes w.r.t. each feature. Define $F_0$ as the set of features with the smallest 1,000 $p$-values

Step 3: For i = 1, 2, ..., K-1, do

Step 4: Cluster the features in $F_{i-1}$ using $D_{train}$ to create a partition, $C_i = \{C_{i1}, C_{i2}, ..., C_{iM_i}\}$

Step 5: For each $C_{ij} \in C_i$, j = 1, 2, ..., $M_i$, do

Step 6: Call $Score\left(C_{ij}, D_{train}, s, t\right)$ to assign a score to each cluster $C_{ij}$

Step 7: End for

Step 8: Rank clusters in descending order and delete the clusters with the lowest $M_i - M_{i+1}$ scores. At the end of this step, the number of clusters in $C_i$ will be $M_{i+1}$

Step 9: For each $C_{ij} \in C_i$, j = 1, 2, ..., $M_{i+1}$, do

Step 10: Call $Importance\left(C_{ij}, D_{train}, s\right)$ to compute an importance score for each feature in cluster $C_{ij}$

Step 11: Remove the $f$% features with the lowest importance score in cluster $C_{ij}$

Step 12: End for

Step 13: Gather the remaining features in all clusters into $F^* = \cup_{l=1}^{M_{i+1}} \{C_{il}\}$

Step 14: Update $D_{train}$ and $D_{test}$ to have the same features as $F^*$

Step 15: Compute performance metrics using $D_{train}$ and $D_{test}$

Step 16: Define $F_{i+1} = F^*$

Step 17: End for

---

**Scoring (S) step:** Each subdataset is assigned a score using a supervised learning algorithm such as support vector machine (SVM) or random forest (RF) (here, we use RF). The subdataset is randomly partitioned into training (70%) and test (30%) sets, where the training data is used for learning the model, and the test data is used for validation. This process is repeated $t$ times and the average classification accuracy serves as the score for the subdataset. Stratified random sampling is applied during partitioning to guarantee that the training and test sets have nearly the same proportion of class labels as the original dataset. In summary, scoring a subdataset is accomplished through the randomized stratified t-fold cross validation (*Prusty, Patnaik & Dash, 2022*). The scoring step is depicted in Fig. 3. It is worth noting that any accuracy index can be used for scoring. Once subdatasets are scored, they are ranked in descending order, and $M_i - M_{i+1}$ clusters with the lowest scores are eliminated as stated before. Eliminating a subdataset means that all features in it are eliminated.

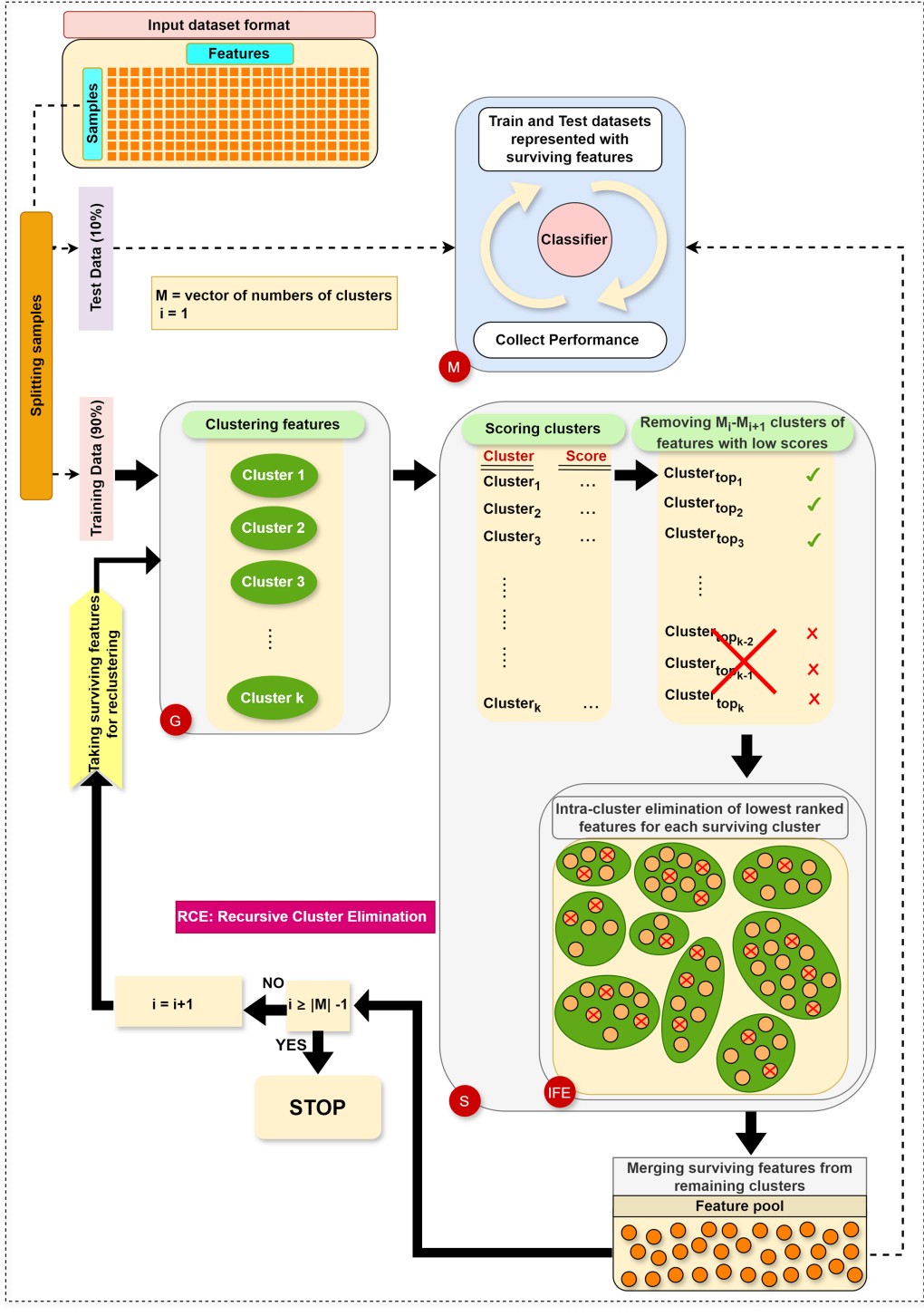

**Figure 1 The workflow of the proposed approach.**

**Internal feature elimination (IFE) step:** Our algorithm eliminates not only low-scoring clusters but also weakly-scoring features within surviving feature clusters. Feature scores are mainly weights or coefficients estimated by a classification algorithm (in our method,

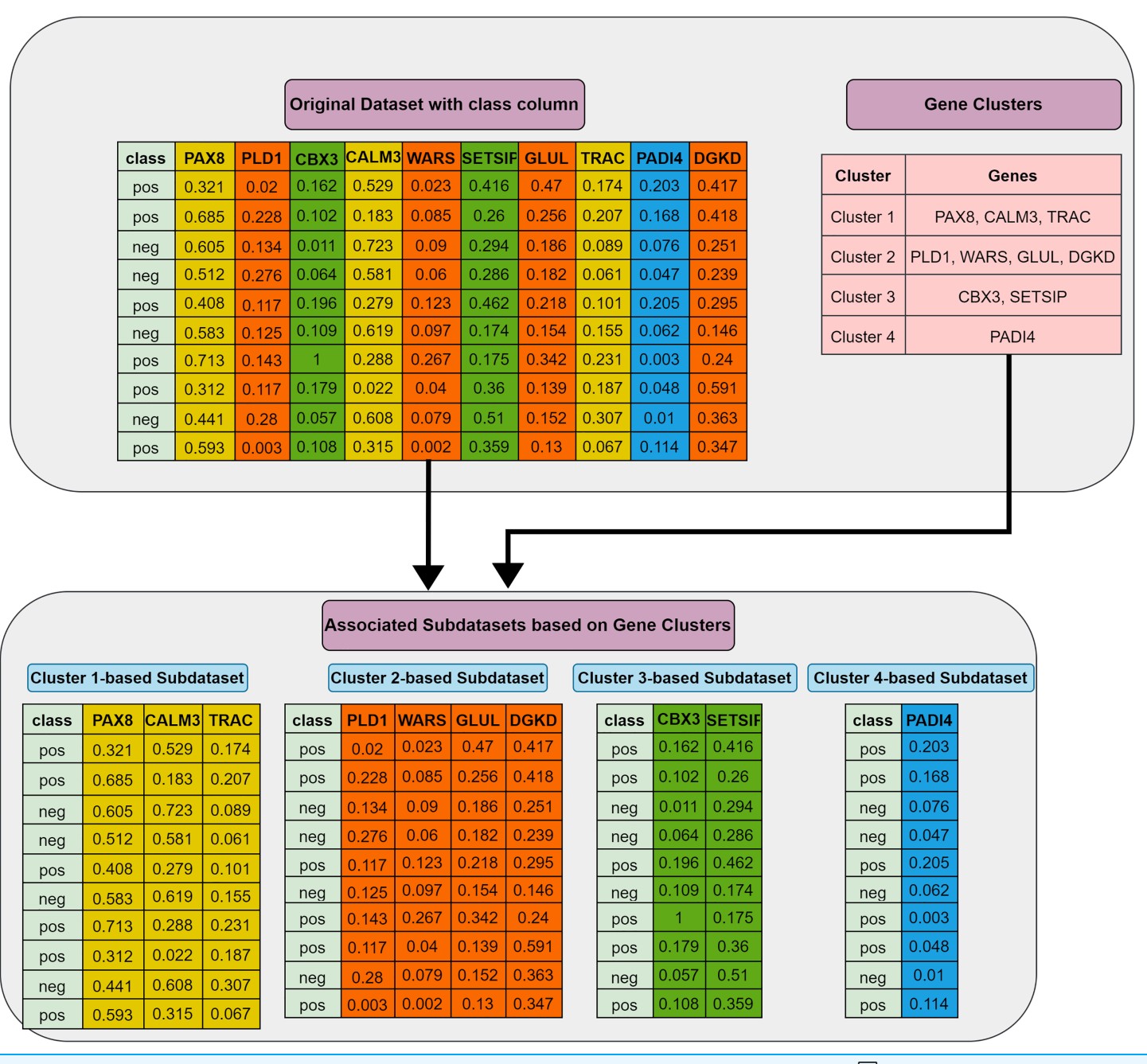

**Figure 2** Extraction of subdatasets based on gene clusters.               

RF). The rate of feature removal is specified in advance. IFE plays a crucial role by further reducing the number of features at each cluster elimination step. This novel process facilitates the exclusion of irrelevant or redundant features, which is, in turn, expected to enhance the generalizability of the model and obtain the same (or better) performance with a smaller number of features. In addition, IFE allows the most important clusters to retain a minimum number of features at the end.

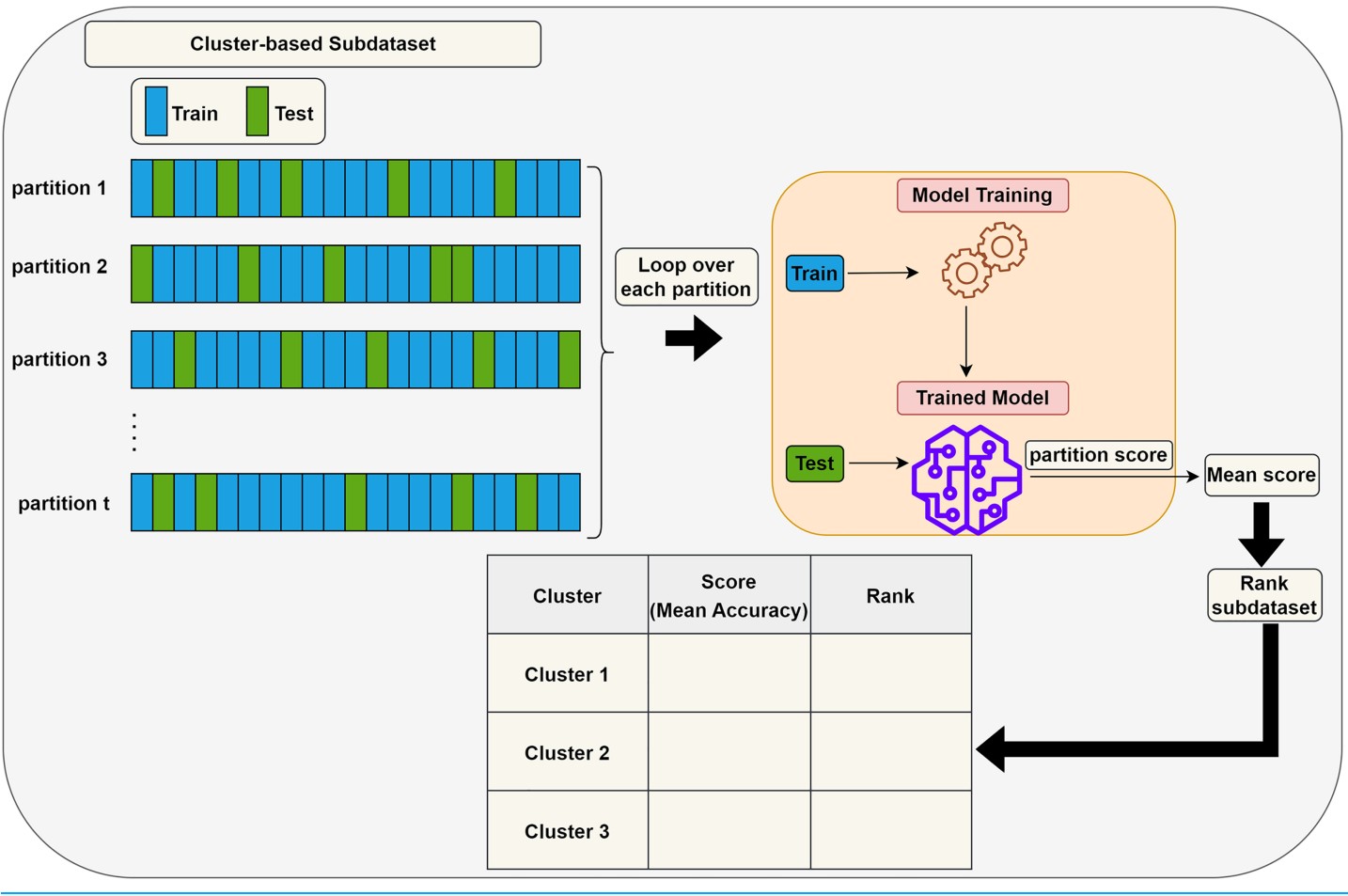

**Figure 3 Assigning a score to a subdataset.**

**Modeling (M) step:** Following the removal of clusters with low scores and intra-cluster elimination of low-scoring features in the remaining clusters, the retained features from the surviving clusters are pooled. Features in this pool are then exploited to construct the training and test datasets, as explained in the G step. Subsequently, an RF model is trained on the training dataset and assessed on the test dataset. At each elimination step, *i.e.*, for each reduced number of clusters, various performance statistics are recorded. Ultimately, we obtain performance results for multiple cluster numbers, and the algorithm terminates after gathering performance results for the final cluster number in $M$.

Several quantitative metrics, including accuracy, sensitivity, specificity, precision, and F-measure, were computed to evaluate the model performance using the formulations below:

$\text{Accuracy} = (\text{TP} + \text{TN})/(\text{TP} + \text{FP} + \text{FN} + \text{TN}),$
$\text{Sensitivity} = \text{TP}/(\text{TP} + \text{FN})$
$\text{Specificity} = \text{TN}/(\text{TN} + \text{FP})$
$\text{Precision} = \text{TP}/(\text{TP} + \text{FP})$
$\text{F-measure} = 2\text{TP}/(2\text{TP} + \text{FP} + \text{FN})$

where TP, FP, TN, and FN refer to the number of correctly predicted positive samples, incorrectly predicted negative samples, correctly predicted negative samples, and incorrectly predicted positive samples, respectively. Furthermore, area under the curve (AUC) is a performance measurement that evaluates the ability of a model to discriminate between two classes, such as the positive class (existence of a disease) and the negative class (lack of a disease). Besides, Cohen's kappa score is a metric that quantifies the degree of agreement between two raters and evaluates the effectiveness of the machine learning model (*McHugh, 2012*). All these metrics are calculated for each cluster number in our approach.

In addition to the aforementioned performance metrics, our method generates a ranked list of features in descending order that persisted successfully through the elimination process based on the number of clusters they survived.

## RESULTS AND DISCUSSION

This section presents the performance of the proposed RCE-IFE approach. We conducted experiments to compare RCE-IFE with SVM-RCE using several types of datasets. We also compared RCE-IFE with some traditional FS methods using various classifiers. The proposed approach was further tested on four publicly available cancer-related datasets and compared with Multi-stage algorithm (*Du et al., 2013*), a method that bears similarities to RCE-IFE. We also compared RCE-IFE with biological domain knowledge-based G-S-M tools, including mirGediNET (*Qumsiyeh, Salah & Yousef, 2023*), 3Mint (*Unlu Yazici et al., 2023*), and miRcorrNet (*Yousef et al., 2021*); and manifested the biological relevancy of the high-scoring features selected by RCE-IFE. Lastly, the consistency of the proposed algorithm in terms of selected features is verified *via* comparing the selected features list among multiple runs.

### Experimental setup

All methods were repeated 100 times to provide stability in results. To ensure balance in datasets, we applied undersampling to remove the samples from the majority class while preserving all of the samples in the minority class. In the cluster scoring step, we applied 10-fold cross validation (*i.e.*, t = 10). For RCE-IFE, we considered the results of two clusters among various cluster numbers. The values presented in our analyses are the average results of 100 repetitions. In RCE-IFE, the rate of features to be removed from a surviving cluster is set to be 10% (*i.e.*, f = 10) if the cluster contains more than five features. AUC is used to evaluate the classifier performance, while the accuracy metric is employed for comparison with Multi-stage algorithm (*Du et al., 2013*), in the Comparative performance evaluation of RCE-IFE with Multi-stage algorithm subsection.

We performed our analyses on Knime (*Berthold et al., 2009*), an open-source software, due to its simplicity and support for graphical representations. In addition, Knime is an extremely integrative tool that allows the integration of different scripts in many coding languages. All experiments were conducted on an Intel Core i9-9900 with 64 GB of RAM.

**Table 4 Comparative performance evaluation of RCE-IFE with SVM-RCE: mean AUC and mean size of the feature subsets on gene expression, miRNA, and methylation datasets.**

| | AUC | | | #Features | | |
|---|---|---|---|---|---|---|
| | SVM-RCE | RCE-IFE-SVM | RCE-IFE | SVM-RCE | RCE-IFE-SVM | RCE-IFE |
| GDS1962 | 0.98 ± 0.05 | 0.99 ± 0.03 | 0.98 ± 0.05 | 76.39 ± 30.23 | 33.93 ± 14.64 | 28.52 ± 10.76 |
| GDS2519 | 0.50 ± 0.18 | 0.56 ± 0.17 | 0.55 ± 0.18 | 46.05 ± 18.44 | 32.69 ± 10.58 | 30.99 ± 9.37 |
| GDS2547 | 0.80 ± 0.11 | 0.83 ± 0.10 | 0.80 ± 0.11 | 109.05 ± 38.84 | 67.84 ± 16.57 | 51.19 ± 16.76 |
| GDS2609 | 0.97 ± 0.14 | 0.97 ± 0.14 | 0.99 ± 0.07 | 59.03 ± 25.06 | 26.34 ± 8.96 | 19.99 ± 7.21 |
| GDS3268 | 0.80 ± 0.12 | 0.78 ± 0.12 | 0.76 ± 0.13 | 99.40 ± 40.34 | 41.44 ± 14.29 | 40.52 ± 14.45 |
| GDS3646 | 0.74 ± 0.23 | 0.77 ± 0.22 | 0.67 ± 0.25 | 38.89 ± 23.88 | 22.56 ± 9.70 | 26.44 ± 10.12 |
| GDS3794 | 0.96 ± 0.12 | 0.94 ± 0.14 | 0.93 ± 0.17 | 63.67 ± 24.61 | 25.52 ± 8.08 | 19.25 ± 6.02 |
| GDS3837 | 0.98 ± 0.03 | 0.98 ± 0.04 | 0.98 ± 0.04 | 124.68 ± 40.48 | 62.60 ± 21.21 | 49.82 ± 16.07 |
| GDS3874 | 0.73 ± 0.20 | 0.75 ± 0.20 | 0.86 ± 0.16 | 42.19 ± 16.94 | 21.53 ± 6.32 | 24.94 ± 7.38 |
| GDS3875 | 0.84 ± 0.15 | 0.86 ± 0.14 | 0.83 ± 0.14 | 41.55 ± 14.32 | 22.22 ± 7.14 | 26.26 ± 7.95 |
| GDS3929 | 0.51 ± 0.28 | 0.48 ± 0.31 | 0.50 ± 0.26 | 17.33 ± 7.30 | 7.26 ± 2.42 | 19.82 ± 6.80 |
| GDS4228 | 0.48 ± 0.26 | 0.46 ± 0.27 | 0.80 ± 0.20 | 12.17 ± 6.18 | 6.63 ± 2.68 | 21.20 ± 6.73 |
| GDS4824 | 0.95 ± 0.22 | 0.95 ± 0.22 | 0.90 ± 0.30 | 61.57 ± 24.17 | 23.32 ± 8.39 | 14.38 ± 5.87 |
| GDS5037 | 0.73 ± 0.24 | 0.72 ± 0.22 | 0.75 ± 0.22 | 37.62 ± 14.15 | 19.00 ± 5.55 | 21.99 ± 7.39 |
| GDS5093 | 0.90 ± 0.22 | 0.91 ± 0.22 | 0.90 ± 0.22 | 42.35 ± 23.24 | 15.40 ± 9.09 | 18.15 ± 7.56 |
| GDS5499 | 0.96 ± 0.07 | 0.95 ± 0.09 | 0.92 ± 0.10 | 179.76 ± 45.84 | 69.92 ± 14.77 | 71.34 ± 22.38 |
| GSE157103 | 0.94 ± 0.10 | 0.97 ± 0.07 | 0.93 ± 0.11 | 95.63 ± 35.05 | 42.30 ± 14.18 | 31.76 ± 11.15 |
| TCGA-BLCA.methylation450 | 0.98 ± 0.06 | 0.98 ± 0.05 | 0.97 ± 0.08 | 73.81 ± 28.14 | 37.65 ± 13.28 | 27.15 ± 10.57 |
| TCGA-BLCA.mirna | 0.97 ± 0.08 | 0.98 ± 0.06 | 0.97 ± 0.08 | 109.19 ± 107.69 | 49.86 ± 42.42 | 25.95 ± 22.90 |
| TCGA-BRCA.methylation450 | 0.98 ± 0.05 | 0.98 ± 0.04 | 0.99 ± 0.03 | 144.27 ± 49.87 | 62.33 ± 21.95 | 53.95 ± 18.80 |
| Average | 0.83 ± 0.15 | 0.84 ± 0.14 | 0.85 ± 0.15 | 73.73 ± 30.74 | 34.52 ± 12.61 | 31.18 ± 11.31 |

## Comparative performance evaluation of RCE-IFE with SVM-RCE on gene expression, miRNA, and methylation datasets

In this section, we compare our proposed approach, RCE-IFE, with SVM-RCE in terms of classification performance, the number of selected features, and execution time. We employed two classifiers, *i.e.*, RF and SVM, individually in our approach. In other words, the steps in scoring clusters and features in surviving clusters were achieved independently using RF and SVM. RF is the default classifier in RCE-IFE, and we denote the approach for integration of SVM into RCE-IFE as RCE-IFE-SVM. Table 4 shows the results on 20 publicly available datasets. The first column corresponds to AUC values, followed by the number of features found by each technique. The last row refers to the average AUC and feature size values obtained by each algorithm across different datasets. We can observe that all methods achieve similar AUC values, where SVM-RCE is outperformed by RCE-IFE-SVM and RCE-IFE by 1% and 2%, respectively. However, a significant difference is observed in the number of selected features between SVM-RCE and RCE-IFE methods. RCE-IFE-SVM selects roughly half as many features as SVM-RCE whereas RCE-IFE picks almost two and a half times fewer features than SVM-RCE. Hence, our proposed approach

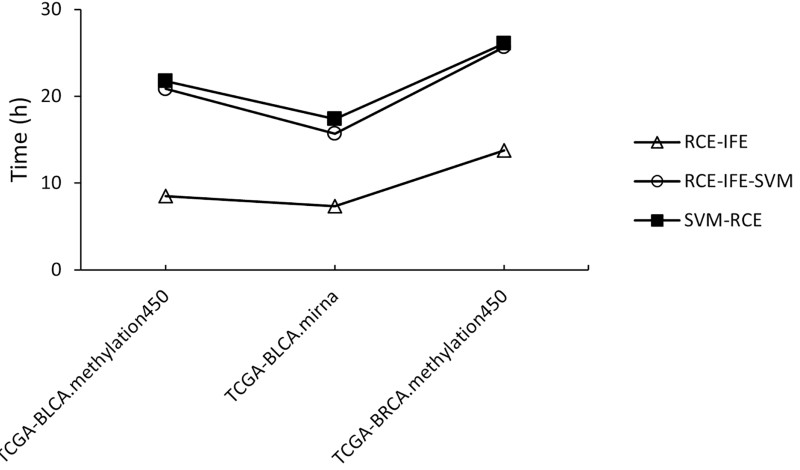

**Figure 4 Running time comparison of RCE-IFE, RCE-IFE-SVM and SVM-RCE on miRNA and methylation datasets.**

**Table 5 Comparative performance evaluation of RCE-IFE with SVM-RCE: mean AUC and mean size of the feature subsets on metagenomics datasets.**

| Dataset | AUC | | | #Features | | |
|---|---|---|---|---|---|---|
| | SVM-RCE | RCE-IFE | % Increase | SVM-RCE | RCE-IFE | % decrease |
| CRC_enzyme | 0.70 ± 0.04 | 0.76 ± 0.04 | 8.6 | 78.51 ± 27.33 | 52.32 ± 17.91 | 33.4 |
| CRC_pathway | 0.68 ± 0.04 | 0.70 ± 0.04 | 2.9 | 83.97 ± 50.93 | 31.60 ± 9.60 | 62.4 |
| CRC_species | 0.67 ± 0.07 | 0.80 ± 0.04 | 19.4 | 96.24 ± 72.55 | 34.65 ± 8.63 | 64.0 |
| CRC_species_II | 0.50 ± 0.20 | 0.82 ± 0.14 | 64.0 | 41.50 ± 47.58 | 16.31 ± 5.84 | 60.7 |
| IBD | 0.82 ± 0.08 | 0.87 ± 0.06 | 6.1 | 89.57 ± 71.94 | 22.12 ± 5.75 | 75.3 |
| IBDMDB | 0.77 ± 0.06 | 0.93 ± 0.04 | 20.8 | 100.07 ± 66.79 | 30.53 ± 8.38 | 69.5 |
| T2D | 0.59 ± 0.12 | 0.66 ± 0.10 | 11.9 | 63.50 ± 58.06 | 16.23 ± 4.13 | 74.4 |
| Average | 0.68 ± 0.09 | 0.79 ± 0.07 | 19.1 | 79.05 ± 56.45 | 29.11 ± 8.61 | 62.8 |

significantly reduces the feature subset size while maintaining the classifier performance. Additional performance metrics can be found in Tables S1–S3.

Regarding running time, RCE-IFE and RCE-IFE-SVM have less time complexity than SVM-RCE, as illustrated for miRNA and methylation datasets in Fig. 4. Note that our proposed approach might seem to have a trade-off between an additional intra-cluster feature elimination step and further feature removal. However, feature removal means dimensionality reduction and outweighs the inclusion of the intra-cluster feature elimination step in favor of shortening execution time. In Fig. 4, we observe that RCE-IFE-SVM shows a moderate reduction in running time compared to SVM-RCE to a certain degree due to the contribution of the intra-cluster elimination step. On the other hand, RCE-IFE has the shortest running time by far. It is also noteworthy that all algorithms achieve the same AUC performance (98%) on average, with different execution times for the datasets.

## Comparative performance evaluation of RCE-IFE with SVM-RCE on metagenomics datasets

To further highlight the dominance of RCE-IFE over SVM-RCE, we tested both methods on seven metagenomics datasets for comparison. The results in Table 5 show that RCE-IFE surpasses SVM-RCE for all datasets in performance and reduced feature subset. Notably, remarkable improvements are observed for CRC and IBDMDB datasets, as seen in the % columns in Table 5. RCE-IFE proves superior to SVM-RCE in both improving classifier performance and reducing feature size. Besides, RCE-IFE generally provides satisfactory results except for T2D, indicating that it is impressive on metagenomics data types. The superiority of RCE-IFE over SVM-RCE in terms of other performance metrics is presented in Table S4.

## Comparative performance evaluation of RCE-IFE with conventional FS methods

This section deals with the comparison of RCE-IFE with several widely used FS algorithms. The results were collected for five gene expression datasets: GDS2547, GDS3268, GDS3646, GDS3875, and GDS5037. We compared seven FS algorithms, which include (1) Minimum Redundancy Maximum Relevance (MRMR) (*Peng, Long & Ding, 2005*), (2) Fast Correlation-Based Filter (FCBF) (*Yu & Liu, 2003*), (3) Information Gain (IG) (*Hall & Smith, 1998*), (4) Conditional Mutual Information Maximization (CMIM) (*Fleuret, 2004*), (5) SelectKBest (SKB), (6) eXtreme Gradient Boosting (XGBoost) (*Chen & Guestrin, 2016*), and (7) SVM-RFE (*Guyon et al., 2002*). In order to evaluate the quality of features obtained by FS methods, eight well-known classification algorithms were applied: (1) Adaboost, (2) Decision Tree (DT), (3) Logitboost, (4) RF, (5) Support Vector Classifier (SVC), (6) Stacking Classifer (base: Logitboost, k-Nearest Neighbour (KNN), final: RF), (7) Stacking Classifier (base: Logitboost, SVC, final: Logistic Regression), and (8) XGBClassifer.

The implementations were carried out through the skfeature and sklearn libraries in python (*Pedregosa et al., 2011*). The number of top genes selected by the FS algorithms was determined according to the average number of genes obtained by RCE-IFE for two clusters. In other words, the number of genes selected by all FS algorithms was kept the same for a fair comparison. Table 6 presents the results for the aforementioned FS algorithms and classifiers with the same number of selected genes. RCE-IFE is superior to the tested FS methods in most cases in terms of AUC (refer to the Average column). While XGBoost with RF reaches the best average performance (79%), the average performance of RCE-IFE (76%) is either quite comparable or dominant over other methods. Out of 49 prediction performances, only 6 show the same or slightly better performance than RCE-IFE. MRMR, FCBF, IG, and CMIM obtain their highest performances with XGB but fall far short of RCE-IFE. However, SKB and XGBoost give competitive results with XGBClassifer. Overall, RCE-IFE outperforms FS algorithms substantially with different classifiers.

**Table 6 Mean AUC values after comparing with popular FS algorithms for the same number of selected genes.**

| FS type | Classifier | # Selected genes 51 GDS2547 | 41 GDS3268 | 26 GDS3646 | 26 GDS3875 | 22 GDS5037 | Average |
|---|---|---|---|---|---|---|---|
| MRMR | Adaboost | 0.49 ± 0.15 | 0.47 ± 0.16 | 0.52 ± 0.28 | 0.46 ± 0.32 | 0.45 ± 0.32 | 0.48 ± 0.25 |
| | DT | 0.49 ± 0.10 | 0.51 ± 0.12 | 0.50 ± 0.20 | 0.48 ± 0.21 | 0.50 ± 0.20 | 0.50 ± 0.17 |
| | Logitboost | 0.49 ± 0.15 | 0.48 ± 0.15 | 0.51 ± 0.25 | 0.47 ± 0.29 | 0.44 ± 0.32 | 0.48 ± 0.23 |
| | RF | 0.49 ± 0.15 | 0.50 ± 0.16 | 0.48 ± 0.29 | 0.44 ± 0.26 | 0.45 ± 0.26 | 0.47 ± 0.22 |
| | SVC | 0.50 ± 0.17 | 0.49 ± 0.15 | 0.59 ± 0.27 | 0.55 ± 0.32 | 0.49 ± 0.31 | 0.52 ± 0.24 |
| | Stacking (Logitboost + KNN) | 0.52 ± 0.15 | 0.49 ± 0.16 | 0.47 ± 0.28 | 0.52 ± 0.30 | 0.56 ± 0.31 | 0.51 ± 0.24 |
| | Stacking (Logitboost + SVC) | 0.51 ± 0.14 | 0.50 ± 0.16 | 0.50 ± 0.27 | 0.53 ± 0.30 | 0.54 ± 0.31 | 0.52 ± 0.24 |
| | XGBClassifier | 0.52 ± 0.14 | 0.75 ± 0.12 | 0.58 ± 0.26 | 0.46 ± 0.27 | 0.65 ± 0.29 | 0.59 ± 0.22 |
| FCBF | Adaboost | 0.61 ± 0.17 | 0.50 ± 0.14 | 0.48 ± 0.32 | 0.53 ± 0.28 | 0.43 ± 0.30 | 0.51 ± 0.24 |
| | DT | 0.57 ± 0.11 | 0.50 ± 0.12 | 0.47 ± 0.21 | 0.52 ± 0.21 | 0.48 ± 0.22 | 0.51 ± 0.17 |
| | Logitboost | 0.60 ± 0.16 | 0.49 ± 0.14 | 0.49 ± 0.33 | 0.50 ± 0.29 | 0.48 ± 0.31 | 0.51 ± 0.25 |
| | RF | 0.64 ± 0.15 | 0.49 ± 0.15 | 0.49 ± 0.30 | 0.51 ± 0.31 | 0.42 ± 0.31 | 0.51 ± 0.24 |
| | SVC | 0.57 ± 0.19 | 0.50 ± 0.16 | 0.52 ± 0.30 | 0.52 ± 0.29 | 0.57 ± 0.26 | 0.54 ± 0.24 |
| | Stacking (Logitboost + KNN) | 0.62 ± 0.16 | 0.49 ± 0.17 | 0.46 ± 0.30 | 0.48 ± 0.28 | 0.43 ± 0.31 | 0.50 ± 0.24 |
| | Stacking (SVC + KNN) | 0.67 ± 0.14 | 0.52 ± 0.14 | 0.50 ± 0.32 | 0.48 ± 0.30 | 0.43 ± 0.30 | 0.52 ± 0.24 |
| | XGBClassifier | 0.64 ±0.18 | 0.62 ± 0.15 | 0.51 ± 0.24 | 0.34 ± 0.28 | 0.65 ± 0.31 | 0.55 ± 0.23 |
| IG | Adaboost | 0.77 ± 0.11 | 0.70 ± 0.14 | 0.53 ± 0.28 | 0.54 ± 0.28 | 0.73 ± 0.27 | 0.65 ± 0.22 |
| | DT | 0.66 ± 0.12 | 0.60 ± 0.14 | 0.50 ± 0.25 | 0.51 ± 0.21 | 0.68 ± 0.22 | 0.59 ± 0.19 |
| | Logitboost | 0.79 ± 0.12 | 0.72 ± 0.12 | 0.53 ± 0.28 | 0.50 ± 0.30 | 0.74 ± 0.25 | 0.66 ± 0.21 |
| | RF | 0.79 ± 0.11 | 0.71 ± 0.13 | 0.53 ± 0.26 | 0.55 ± 0.25 | 0.79 ± 0.23 | 0.67 ± 0.20 |
| | SVC | 0.59 ± 0.26 | 0.33 ± 0.14 | 0.49 ± 0.26 | 0.46 ± 0.30 | 0.31 ± 0.25 | 0.44 ± 0.24 |
| | Stacking (Logitboost + KNN) | 0.73 ± 0.12 | 0.57 ± 0.15 | 0.46 ± 0.26 | 0.53 ± 0.27 | 0.65 ± 0.29 | 0.59 ± 0.22 |
| | Stacking (SVC + KNN) | 0.76 ± 0.11 | 0.62 ± 0.19 | 0.47 ± 0.26 | 0.57 ± 0.29 | 0.61 ± 0.32 | 0.61 ± 0.23 |
| | XGBClassifier | 0.80 ± 0.11 | 0.74 ± 0.12 | 0.51 ± 0.27 | 0.51 ± 0.30 | 0.78 ± 0.24 | 0.67 ± 0.21 |
| CMIM | Adaboost | 0.51 ± 0.17 | 0.58 ± 0.15 | 0.51 ± 0.26 | 0.71 ± 0.26 | 0.83 ± 0.24 | 0.63 ± 0.22 |
| | DT | 0.53 ± 0.13 | 0.57 ± 0.12 | 0.52 ± 0.21 | 0.79 ± 0.18 | 0.73 ± 0.20 | 0.63 ± 0.17 |
| | Logitboost | 0.54 ± 0.17 | 0.58 ± 0.14 | 0.50 ± 0.27 | 0.79 ± 0.24 | 0.81 ± 0.24 | 0.64 ± 0.21 |
| | RF | 0.58 ± 0.17 | 0.56 ± 0.14 | 0.54 ± 0.27 | 0.84 ± 0.23 | 0.74 ± 0.27 | 0.65 ± 0.22 |
| | SVC | 0.46 ± 0.17 | 0.45 ± 0.14 | 0.50 ± 0.26 | 0.22 ± 0.27 | 0.45 ± 0.27 | 0.42 ± 0.22 |
| | Stacking (Logitboost + KNN) | 0.50 ± 0.17 | 0.47 ± 0.15 | 0.49 ± 0.28 | 0.64 ± 0.26 | 0.57 ± 0.32 | 0.53 ± 0.24 |
| | Stacking (SVC + KNN) | 0.56 ± 0.17 | 0.43 ± 0.13 | 0.51 ± 0.29 | 0.64 ± 0.29 | 0.59 ± 0.30 | 0.55 ± 0.24 |
| | XGBClassifier | 0.56 ± 0.15 | 0.64 ± 0.13 | 0.44 ± 0.27 | 0.80 ± 0.24 | 0.84 ± 0.21 | 0.66 ± 0.20 |
| SKB | Adaboost | 0.80 ± 0.11 | 0.79 ± 0.10 | 0.70 ± 0.25 | 0.73 ± 0.23 | 0.72 ± 0.26 | 0.75 ± 0.19 |
| | DT | 0.67 ± 0.12 | 0.63 ± 0.13 | 0.61 ± 0.22 | 0.57 ± 0.18 | 0.60 ± 0.18 | 0.62 ± 0.17 |
| | Logitboost | 0.79 ± 0.11 | 0.80 ± 0.10 | 0.75 ± 0.24 | 0.73 ± 0.24 | 0.79 ± 0.24 | 0.77 ± 0.19 |
| | RF | 0.81 ± 0.11 | 0.80 ± 0.10 | 0.74 ± 0.24 | 0.78 ± 0.20 | 0.78 ± 0.23 | 0.78 ± 0.18 |
| | SVC | 0.79 ± 0.12 | 0.23 ± 0.12 | 0.25 ± 0.25 | 0.29 ± 0.22 | 0.26 ± 0.24 | 0.36 ± 0.19 |
| | Stacking (Logitboost + KNN) | 0.72 ± 0.12 | 0.74 ± 0.13 | 0.74 ± 0.24 | 0.68 ± 0.23 | 0.76 ± 0.24 | 0.73 ± 0.19 |
| | Stacking (SVC + KNN) | 0.81 ± 0.10 | 0.80 ± 0.11 | 0.76 ± 0.25 | 0.66 ± 0.26 | 0.73 ± 0.25 | 0.75 ± 0.19 |
| | XGBClassifier | 0.81 ± 0.10 | 0.81 ± 0.12 | 0.70 ± 0.25 | 0.71 ± 0.22 | 0.76 ± 0.25 | 0.76 ± 0.19 |

| FS type | Classifier | # Selected genes | | | | | |
| | | 51 | 41 | 26 | 26 | 22 | |
| | | GDS2547 | GDS3268 | GDS3646 | GDS3875 | GDS5037 | Average |
|---|---|---|---|---|---|---|---|
| XGBoost | Adaboost | 0.79 ± 0.12 | 0.84 ± 0.10 | 0.64 ± 0.28 | 0.77 ± 0.20 | 0.67 ± 0.27 | 0.74 ± 0.19 |
| | DT | 0.64 ± 0.13 | 0.67 ± 0.11 | 0.55 ± 0.23 | 0.69 ± 0.20 | 0.62 ± 0.26 | 0.63 ± 0.20 |
| | Logitboost | 0.80 ± 0.11 | 0.86 ± 0.09 | 0.61 ± 0.28 | 0.80 ± 0.22 | 0.75 ± 0.26 | 0.76 ± 0.19 |
| | RF | 0.82 ± 0.11 | 0.85 ± 0.10 | 0.68 ± 0.26 | 0.82 ± 0.21 | 0.77 ± 0.25 | 0.79 ± 0.19 |
| | SVC | 0.78 ± 0.11 | 0.19 ± 0.11 | 0.38 ± 0.29 | 0.25 ± 0.23 | 0.27 ± 0.25 | 0.37 ± 0.20 |
| | Stacking (Logitboost + KNN) | 0.78 ± 0.11 | 0.83 ± 0.10 | 0.62 ± 0.26 | 0.77 ± 0.21 | 0.74 ± 0.23 | 0.75 ± 0.18 |
| | Stacking (SVC + KNN) | 0.80 ± 0.11 | 0.80 ± 0.11 | 0.64 ± 0.27 | 0.75 ± 0.22 | 0.74 ± 0.29 | 0.75 ± 0.20 |
| | XGBClassifier | 0.81 ± 0.12 | 0.85 ± 0.10 | 0.63 ± 0.27 | 0.83 ± 0.20 | 0.76 ± 0.26 | 0.78 ± 0.19 |
| SVM-RFE | SVM | 0.78 ± 0.08 | 0.83 ± 0.07 | 0.74 ± 0.16 | 0.70 ± 0.17 | 0.72 ± 0.19 | 0.75 ± 0.13 |
| RCE-IFE | RF | 0.80 ± 0.11 | 0.76 ± 0.13 | 0.67 ± 0.25 | 0.83 ± 0.14 | 0.75 ± 0.22 | 0.76 ± 0.17 |

**Table 7 Comparative performance evaluation of RCE-IFE with Multi-stage: Mean classification accuracies of RCE-IFE and Multi-stage.**

| Datasets | Multi-stage | RCE-IFE |
|---|---|---|
| Breast | 82.84 | 80.89 ‖ 44 |
| DLBCL | 88.74 | 87.67 ‖ 34 |
| Leukemia | 97.59 | 96.33 ‖ 70 |
| Prostate | 93.45 | 92.18 ‖ 38 |
| Average | 90.66 | 89.27 ‖ 46.5 |

**Note:**
On the RCE-IFE column, the numbers on the right of the "‖" symbol indicate the average number of selected genes.

## Comparative performance evaluation of RCE-IFE with multi-stage algorithm

In this section, RCE-IFE is tested with four widely used cancer-related datasets that are readily available. We compared RCE-IFE with Multi-stage algorithm (*Du et al., 2013*) due to its similarity to our approach. While both methods share some common steps, the way these steps are performed differs, as Multi-stage employs SVM-RFE and adopts a distinct strategy for cluster elimination. Table 7 presents the performance metrics obtained for RCE-IFE and Multi-stage on four datasets. The metrics for Multi-stage are extracted from the original study, where performance was reported as the average accuracy of the top 60 genes. For RCE-IFE, the numbers on the right of the "‖" symbol refer to the average number of selected genes. As shown in Table 7, RCE-IFE achieves accuracy rates comparable to those of Multi-stage, with insignificant differences. Moreover, except for the Leukemia dataset, the number of genes selected by RCE-IFE is considerably below 60 in all cases. These findings imply that RCE-IFE has a very competitive performance in terms of accuracy measure and outperforms Multi-stage in terms of gene reduction.

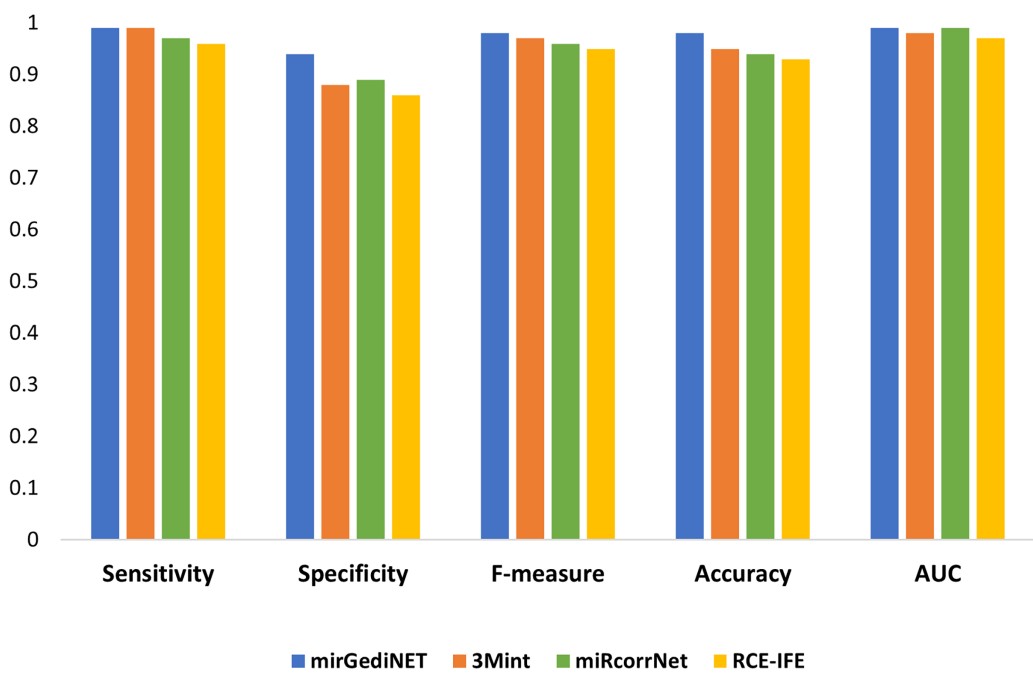

**Figure 5** Comparative performance evaluation of RCE-IFE with three biological domain knowledge-based G-S-M tools.

## Comparative performance evaluation of RCE-IFE with other biological domain knowledge-based G-S-M tools

We have comparatively evaluated RCE-IFE with three biological domain knowledge-based G-S-M tools, *i.e.*, mirGediNET (*Qumsiyeh, Salah & Yousef, 2023*), 3Mint (*Unlu Yazici et al., 2023*), and miRcorrNet (*Yousef et al., 2021*), in terms of several performance metrics. These tools integrate biological knowledge into feature grouping and score the feature groups using a classifier. For comparison purposes, we employed the TCGA-BRCA dataset used in the aforementioned tools, available on the Genomic Data Commons (GDC) repository hosted by the National Cancer Institute (NCI). This dataset is a type of miRNA expression with the reads mapped to GRCh38 and downloaded from the UCSC Xena repository (https://xenabrowser.net/datapages/) (*Goldman et al., 2020*). For this dataset, tumor samples were filtered such that Luminal A and Luminal B subtypes (248 ER+/PR +/PR-samples) were considered positive (LumAB), while Her2-enriched and Basal-like subtypes (124 ER-/PR-samples) were considered negative (Her2Basal).

We selected the best performance metrics for each tool based on their highest AUC values for the TCGA-BRCA molecular subtype dataset and compared them with RCE-IFE. mirGediNET, 3Mint, and RCE-IFE obtain these AUC values *via* selecting a similar number of features, *i.e.*, 9.6, 13.6, and 12.4 features, respectively, when averaged over 100 iterations. In contrast, miRcorrNet selects a larger number of features, *i.e.*, 38.2 on average. Figure 5 plots several performance metrics obtained with these four tools when tested on the TCGA-BRCA molecular subtype dataset. As apparent in Fig. 5, the performance measurements of RCE-IFE, miRcorrNet, 3Mint, and mirGediNET on the TCGA-BRCA molecular subtype dataset are close, implying that these tools are comparable. However,

**Table 8 Top five miRNAs selected on the TCGA-BRCA molecular subtype dataset.**

| Rank | Name | Accession ID |
|---|---|---|
| 1 | hsa-mir-190b | MI0005545 |
| 2 | hsa-mir-135b | MI0000810 |
| 3 | hsa-mir-18a | MI0000072 |
| 4 | hsa-mir-505 | MI0003190 |
| 5 | hsa-mir-934 | MI0005756 |

the outcome is different for each tool as each has its own benefit and is developed to uncover the significant groups based on specific biological knowledge.

## Biological validation of the RCE-IFE findings

In this section, we analyze the biological relevancy and significance of the selected miRNAs for the TCGA-BRCA dataset introduced in the previous section. Breast cancer is the most commonly diagnosed malignancy worldwide and it is the leading cause of cancer-related mortality among women (*Frick et al., 2024*). The top five miRNAs selected by our method on the TCGA-BRCA molecular subtype dataset are listed in Table 8. The top feature, miR-190b, was reported to be upregulated in patients with ER+ breast cancer, and its dysregulation plays a significant role in the initiation and progression of breast cancer (*Dai et al., 2019*). This miRNA was found to suppress breast cancer metastasis by targeting SMAD2 (*Yu et al., 2018*). MiR-135b is a key regulator in breast cancer and promotes tumor growth, invasion, and metastasis (*Hua et al., 2016*). It serves as an oncogene, and due to its effect in modulating critical signaling pathways, it is considered a potential biomarker in breast cancer and a promising target for therapeutic intervention (*Vo et al., 2024*). The expression of miR-18a correlates with ER-breast tumors characterized by a high level of inflammation (*Egeland et al., 2020*). High expression of it is closely linked to basal-like breast cancer (*Jonsdottir et al., 2012*). Downregulation of miR-505 promotes cellular processes such as cell proliferation, migration, and invasion. Moreover, the low expression level of this miRNA is associated with poor prognosis in breast cancer patients (*Wang, Liu & Li, 2019*). MiR-934 expression has a strong association with the overall survival of breast cancer patients and promotes cell metastasis by targeting PTEN (*Lu, Hu & Yang, 2021*). In addition, mir934 inhibition suppresses the migration capability of tumor cells to a certain extent in patients with triple-negative breast cancer (*Contreras-Rodríguez et al., 2023*).

## Consistency of the selected features across different runs

To evaluate the consistency of miRNAs selected by the proposed algorithm, we run RCE-IFE on the TCGA-BRCA molecular subtype dataset three times, where each run consists of 100 iterations. Top selected miRNAs were compared among successive runs. Figure 6 depicts the overlaps of the top (A) five miRNAs, (B) 10 miRNAs, and (C) 20 miRNAs between three different runs. As apparent in Fig. 6, the top five miRNAs are the same miRNAs for all the runs. Among the top 10 miRNAs selected in three different runs, nine

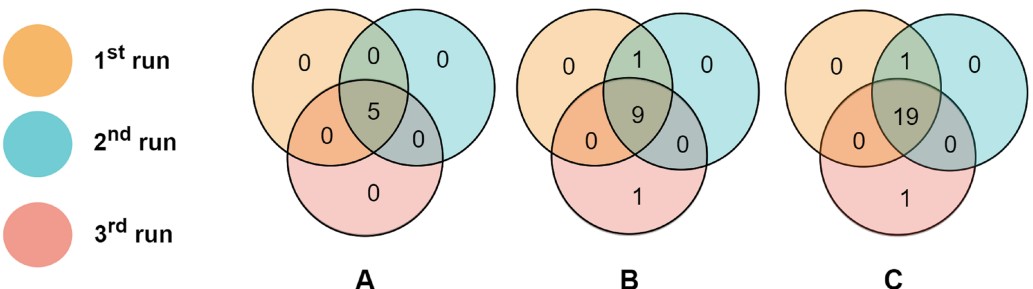

**Figure 6  Overlaps between the top miRNAs selected in different runs.** (A) Top five miRNAs. (B) Top 10 miRNAs. (C) Top 20 miRNAs.

are commonly identified in all three runs, and one miRNA is common among the two runs. Only one miRNA is non-overlapping with any other runs. We encounter a similar situation when the top 20 features are selected in three different runs. A total of 19 miRNAs are commonly detected in all runs, one miRNA is shared between the results of the two runs, and one miRNA is unique to one run. These results reveal the large-scale consistency of the miRNAs selected by the RCE-IFE algorithm across different runs.

## CONCLUSIONS

In this article, we address the challenge of the FS task using a feature grouping-based strategy. We propose RCE-IFE, which involves cluster elimination followed by removing features from surviving clusters to retain non-redundant and strongly relevant features. This approach leads to further dimensionality reduction, improved feature subset quality, enhanced model performance, and reduced computation time. We have conducted different experiments on publicly available datasets from several biological domains to assess the performance of RCE-IFE.

RCE-IFE, RCE-IFE-SVM (SVM adapted version of RCE-IFE), and SVM-RCE yield comparable average AUCs of 0.85, 0.84, and 0.83, respectively, when averaged over different gene expression, miRNA expression, and methylation datasets; however, RCE-IFE achieves this performance *via* selecting the fewest features and the least execution time. When tested on seven different metagenomics datasets, RCE-IFE and SVM-RCE obtain average AUCs of 0.79 and 0.68, respectively. In these experiments on metagenomics dataset, RCE-IFE demonstrates a significant reduction in the size of feature subsets. In addition, RCE-IFE outperforms various popular FS methods, including MRMR, FCBF, IG, CMIM, SKB, and XGBoost, achieving an average AUC of 0.76 on five gene expression datasets. Compared to a similar tool, Multi-stage, RCE-IFE acquires a similar average accuracy rate of 89.27% using a lower number of features on four cancer-related datasets.

Additionally, we show that the performance of RCE-IFE is comparable with other biological domain knowledge-based G-S-M tools (mirGediNET, 3Mint, and miRcorrNet) on the TCGA-BRCA dataset. We verify in scientific literature that all of the identified top five miRNAs (miR-190b, miR-135b, miR-18a, miR-505, and miR-934) play significant roles in disease progression and prognosis, contributing to the elucidation of the molecular mechanisms of breast cancer and the development of treatment strategies. Finally, we

indicate that the proposed algorithm is capable of selecting features with a high degree of consistency across multiple runs. Our findings suggest that the proposed method is powerful in that it provides robust model prediction, achieves a substantial reduction in feature dimension, and ensures both feature relevancy and consistency.

As for future work, we intend to experiment with additional types of datasets from diverse domains, such as text or image. We also plan to implement the selection of a fixed number of representative features from each cluster to further reduce the computational overhead. Lastly, the proposed framework is intended to be adaptable for multiclass classification and multi-label FS problems.

### Funding

This work has been supported by the Zefat Academic College. Burcu Bakir-Gungor's work has been supported by the Abdullah Gul University Support Foundation (AGUV). The funders had no role in study design, data collection and analysis, decision to publish, or preparation of the manuscript.

### Grant Disclosures

The following grant information was disclosed by the authors:
Zefat Academic College.
Abdullah Gul University Support Foundation (AGUV).

### Competing Interests

Burcu Bakir-Gungor is an Academic Editor for Peerj.

### Author Contributions

- Cihan Kuzudisli conceived and designed the experiments, performed the experiments, analyzed the data, performed the computation work, prepared figures and/or tables, authored or reviewed drafts of the article, and approved the final draft.
- Burcu Bakir-Gungor conceived and designed the experiments, performed the experiments, analyzed the data, performed the computation work, prepared figures and/or tables, authored or reviewed drafts of the article, and approved the final draft.
- Bahjat Qaqish performed the experiments, analyzed the data, prepared figures and/or tables, authored or reviewed drafts of the article, and approved the final draft.
- Malik Yousef conceived and designed the experiments, analyzed the data, performed the computation work, prepared figures and/or tables, authored or reviewed drafts of the article, and approved the final draft.

### Data Availability

The gene expression data is available at NCBI GEO: GDS1962, GDS2547, GDS3268, GDS3646, GDS3837, GDS3874, GDS5037, GDS5499, GDS2519, GDS2609, GDS3875, GDS3929, GDS4228, GDS4824, GDS5093, GDS3794, GSE157103.

The TCGA-BLCA.methylation450 and TCGA-BLCA.mirna datasets are available in GDC TCGA Bladder Cancer (BLCA) cohort. The TCGA-BRCA.methylation450 dataset is available at GDC TCGA Breast Cancer (BRCA) cohort. These datasets are available at UCSC Xena repository (https://xenabrowser.net/datapages/).

The CRC_species, CRC_pathway and CRC_enzyme datasets are available in the Supplemental Material in *Beghini et al. (2021)*.

The IBD dataset is available at European Nucleotide Archive (ENA): ERA000116.

The IBDMDB dataset is available in NCBI BioProject: PRJNA398089.

The T2D dataset is available at NCBI: SRA045646 and SRA050230. The CRC_species_II dataset is available at European Nucleotide Archive (ENA): PRJEB6070.

The leukemia dataset is available from https://www.openintro.org/data/index.php?data=golub.

The prostate dataset is available from R package SIS: https://cran.r-project.org/web/packages/SIS/SIS.pdf.

The breast dataset is available in the Breast Cancer (vantVeer 2002) cohort in UCSC Xena repository at https://xenabrowser.net/datapages/.

The DLBCL dataset is available at GitHub: https://github.com/ramhiser/datamicroarray.

The TCGA-BRCA miRNA expression breast cancer dataset is available on the Genomic Data Commons (GDC) repository and downloaded from GDC TCGA Breast Cancer (BRCA) cohort at UCSC Xena repository (https://xenabrowser.net/datapages/).

The code is available at GitHub: https://github.com/malikyousef/RCE-IFE.

The DOI of the tool is available at Zenodo: Malik Yousef, & ckuzudisli. (2024). malikyousef/RCE-IFE: First version (Version v1). Zenodo. https://doi.org/10.5281/zenodo.11167403.

## Supplemental Information

Supplemental information for this article can be found online at http://dx.doi.org/10.7717/peerj-cs.2528#supplemental-information.

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
