# Peer review of "RCE-IFE: recursive cluster elimination with intra-cluster feature elimination"

_PeerJ Computer Science, doi:10.7717/peerj-cs.2528_

## Round 0.1 · original submission · Minor Revisions

In spite of my decision "Minor", I expect that the authors will follow carefully after all comments and suggestions of both reviewers.

**Language Note:** The review process has identified that the English language must be improved. PeerJ can provide language editing services - please contact us at [email protected] for pricing (be sure to provide your manuscript number and title). Alternatively, you should make your own arrangements to improve the language quality and provide details in your response letter. – PeerJ Staff

·

Basic reporting

1. Use Grammarly software for any spelling and grammatical errors, also improve the English writing throughout the paper.
2. In introduction part, the authors have given 8 paragraphs. This should be in 3 to 4 paragraphs, like the 1st paragraphs-the background, 2nd the previous research and major gaps, 3rd should be your contribution and findings, and last should be the structure of the paper.
3. Headings should be properly numbered i.e. Introduction, Materials…
4. Two datasets link are missing in the last paragraph of 2.1.
5. Figures are not cited same. Like in some place it has Figure (Line no. 237) and other pace as Fig. (Line no. 257).
6. Figure 2 is not clear. Use different background colors and increase the font size to make those visible.
7. In table 1, the author has given 20 datasets, which are not in ordered. These should be according to samples or features or dataset numbers. Similarly, in other tables.
8. Caption in Table 4 require to be updated with the proposed method.
9. References should be properly arranged in ascending form.

Experimental design

1. The abstract needs to be improvised. It contains more introduction. Only reflect your contributions and findings in this research according to the different datasets.
2. In line 170, Algorithm 1 presents the pseudocode for this step, which looks incomplete. Similarly, in line 174,, algorithm 2 gives the pseudocode for the feature scoring step, which looks incomplete.
3. Performance evaluation metrics need to be described. Table 4 and 5 includes only ‘Auc’ metric. These tables should include all other classification metrics.
4. In line no. 342, include the table number in that sentence.
5. It will be better if the heading of Table 6 will also display in second page.

Validity of the findings

1. Design a comparison table containing all performance evaluation metrics between your main findings and other previous research in cancer related research.
2. Describe the gaps in previous research and the advantagess of your method over those in discussion section.
3. Improvise the conclusion section. Include your main findings and future perspectives.
4. Add all dataset information to the supplementary materials.

Additional comments

Improve the abstract, introduction, and conclusion sections.

·

Basic reporting

1. The authors developed Recursive Cluster Elimination with Intra-Cluster Feature Elimination
(RCE-IFE), a method that iterates clustering and elimination steps in a supervised context (e.g., with a lable of pre and post). Then, they assess the performance in comparison to couple of other tools on multiple types of biological data.
2. They described their algorithm very well and to the necessary details.
3. K in MK was used without definition in line 161 in 2.2 Proposed Algorithm.

Experimental design

1. Since this algorithm involves many arbitrary parameters (e.g., the rate of feature removal or Mi for each i, or t in t-fold cross validation), it would be more desirable to study the effect of different parameter values.
2. Since most of the algorithms have random factors, they can look into how much overlap they make across different runs of different random factors to show consistency in feature selection.

Validity of the findings

1. Isn't the reason RCE-IEF picks fewer genes than SVM-RCE only because of the low rate of feature selection parameter set that way?
2. Also, isn't the low running time (and time complexity) related to also the the low rate of feature selection parameter set that way?
3. It would be a lot more desirable if they validate the selected features using biological literature not just in terms of the number. Again, you can always reduce the number by setting harsh cutoffs.

---

## Round 0.2 · accepted · Accept

Thank you for carefully following the valuable suggestions of the reviewers. I confirm that your manuscript was substantially improved